# Ancient Homozygosity Segments in West African Djallonké Sheep Inform on the Genomic Impact of Livestock Adaptation to the Environment

**DOI:** 10.3390/ani10071178

**Published:** 2020-07-12

**Authors:** Isabel Álvarez, Iván Fernández, Amadou Traoré, Lucía Pérez-Pardal, Nuria A. Menéndez-Arias, Félix Goyache

**Affiliations:** 1Servicio Regional de Investigación y Desarrollo Agroalimentario, E-33394 Gijón, Spain; ialvarez@serida.org (I.Á.); ifernandez@serida.org (I.F.); nuriamendezarias@hotmail.com (N.A.M.-A.); 2Institut de l’Environnement et des RecherchesAgricoles (INERA), 8645 Ouagadougou BP, Burkina Faso; traore_pa@yahoo.fr; 3CIBIO-InBIO, Universidade do Porto, 4485-661 Vairão, Portugal; luciaperez131281@hotmail.com

**Keywords:** West African dwarf sheep, domestication, adaptation, runs of homozygosity, identity-by-descent

## Abstract

**Simple Summary:**

Adaptation to the challenging environment of humid Sub-Saharan West Africa is hypothesized to cause an effect on the sheep genome. This can be used to identify genomic areas of importance to adaptation to changing environments. Djallonké sheep are small-sized, hair-coated trypano tolerant animals resulting from a unique process of natural adaptation. Here we identify ancient homozygous stretches, considered homozygous-by-descent, on the Djallonké sheep genome. Such genomic segments were assumed to be inherited from ancestors living during the historical time when sheep entered into West Africa and, therefore, are considered informative on the effect of human-mediated selection on the sheep genome. The genomic areas identified were involved in homeostasis and coagulation, innate immunity, defense against infections, white blood cells proliferation and migration, parasite resistance, and response to stress.

**Abstract:**

A sample of Burkina Faso Djallonké (West African Dwarf) sheep was analyzed to identify stretches of homozygous segments (runs of homozygosity; ROH) overlapping with ancient homozygosity-by-descent (HBD) segments. HBD segments were considered ancient if they were likely to be inherited from ancestors living from 1024 to 2048 generations ago, roughly coinciding with the time in which sheep entered into West Africa. It is hypothesized that such homozygous segments can inform on the effect of the sheep genome of human-mediated selection for adaptation to this harsh environment. PLINK analyses allowed to identify a total of 510 ROH segments in 127 different individuals that could be summarized into 124 different ROH. A total of 32,968 HBD segments were identified on 119 individuals using the software ZooRoH. HBD segments inherited from ancestors living 1024 and 2048 generations ago were identified on 61 individuals. The overlap between consensus ROH identified using PLINK and HBD fragments putatively assigned to generations 1024 and 2048 gave 108 genomic areas located on 17 different ovine chromosomes which were considered candidate regions for gene-annotation enrichment analyses. Functional annotation allowed to identify six statistically significant functional clusters involving 50 candidate genes. Cluster 1 was involved in homeostasis and coagulation; functional clusters 2, 3, and 6 were associated to innate immunity, defense against infections, and white blood cells proliferation and migration, respectively; cluster 4 was involved in parasite resistance; and functional cluster 5, formed by 20 genes, was involved in response to stress. The current analysis confirms the importance of genomic areas associated to immunity, disease resistance, and response to stress for adaptation of sheep to the challenging environment of humid Sub-Saharan West Africa.

## 1. Introduction

Although domestication of sheep was carried out in the Fertile Crescent [1] about 11,000 years before present (yBP), these livestock only entered into West Africa by 3700 yBP [2]. This contrasts with the surrounding geographical regions: sheep reached central Nile valley and central Sahara about 6000 yBP [2]. Archaeologists suggest that this delay in the formation of pastoralist societies in biogeographic zones south of the Sahel stemmed from new animal diseases encountered by pastoral colonists, such as trypanosomiasis [3]. Djallonké (West African Dwarf) sheep [4,5] are a small-sized hair-coated sheep population considered resilient to trypanosome challenge [6].

Human-mediated selection for adaptation to harsh environments may shape the livestock genome. Although most analyses dealing with this issue have been performed via identifying selection sweeps on the genome (see [7] for a recent review), an alternative approach would consist of the identification of stretches of homozygous segments in the genome, usually referred to as runs of homozygosity (ROH) [8,9,10]. Genomic regions subjected to selection frequently show reduced nucleotide diversity and increased homozygosity around the selected loci [9]. Therefore, the abundance, length, and genomic distribution of ROH have been often used as a valuable source of information about the demographic history of species [8,11]. However, there is no consensus on the use of ROH to ascertain evolutionary genomic events. ROH are strongly influenced by recent demographic events such as population decline and unbalanced paternal contributions [8,11] or the influence of artificial selection programs [12], usually causing long ROH segments in the genome [8]. Furthermore, other factors such as recombination rate and GC content can affect genomic ROH size, abundance, and distribution [11].

ROH segments have been shown to be informative in population genetics to estimate inbreeding or effective size [13,14] and to assess inbreeding depression [15]. However, the use of ROH as markers for the identification of genomic areas potentially subjected to non-recent evolutionary selective events is not straightforward. It requires to ensure that homozygous segments were not caused by recent demographic events and have been inherited from old ancestors. Recent approaches [16,17] assessed the number of generations connecting genomic homozygous segments to the time of living of the most recent common ancestor. These segments are considered homozygous-by-descent (HBD). Due to recombination, HBD segment lengths are inversely related with their time of origin (in generations): the older the origin of inbreeding, the shorter (on average) stretches of observed homozygous markers.

In this research, a sample of Burkina Faso Djallonké sheep was typed using medium-density DNA chips. Genomic data were analyzed to identify ancient HBD segments. Such segments, putatively spanning genes involved in the adaptation of sheep to new environmental conditions, were subject to functional characterization to assess the genomic impact of adaptation of domesticated sheep to humid sub-Saharan West Africa.

## 2. Materials and Methods

### 2.1. Data Available

We explicitly state that the use of samples in the current research follows the ‘three-R principles’: R1: replacement, as Djallonké sheep have not been characterized thus far at the homozygous stretches level, and the current study is thus not redundant with other data; R2: reduction, as the samples were originally collected for previous projects (by Joint FAO/IAEA Coordinated Research Project D3.10.26 by Fond Compétitif National of Burkina Faso CNRST/PPAAO) being shared to maximize their contribution to research and knowledge production; and R3: refinement, as the samples were obtained using standard veterinary procedures in full respect of animal welfare and minimizing stress. Blood and hair root samples used here were collected by veterinary practitioners with the permission, and in the presence of, the owners. For this reason, permission from the Ethics Committee for Health Research in Burkina Faso (Joint Order 2004-147/MS/MESSE of May 11, 2004) was not required.

Data, comprising 184 DNA samples from Djallonké lambs (64 males and 120 females) were previously analyzed using microsatellites and Single Nucleoride Polimorphism (SNP) data [7,18]. Samples were obtained during a field trial for the assessment of gastrointestinal parasite resistance [18,19,20] and DNA extracted using standard procedures [21] (see previous references for full details of sampling). Briefly, 166 individuals were sampled in Mangodara (latitude 9°53′59.99″ N; longitude 4°20′59.99″ W; Comoé province) and the other 18 (9 males and 9 females) in Dédougou (latitude 12°27′48.17″ N; longitude 3°27′38.7″ W; Mouhoun province). Both provinces are located in southern Burkina Faso within the Sudan-Guinea Savannah humid environmental region (tsetse challenged) of the country. The Sudan-Guinea Savannah environmental region has annual rainfall higher than 900 mm with a predominance of woodlands and savannahs. Precipitations in the sampling areas range from 1000 to 1200 mm per year and temperatures vary from 19 °C to 36 °C. Agro-ecological constraints affecting tsetse fly distribution and trypanosome challenge in Burkina Faso have been widely described and discussed in previous works [22].

Genotypes [7] were obtained using the Ovine 50 K SNP BeadChip and the software GenomeStudio (Illumina Inc., San Diego, CA, USA) fitting a GenCall score cutoff of 0.15 and average sample call rate of 99%. Standard .ped and .map files were subject to quality control using the program PLINK V 1.9 [23]. All unmapped SNPs, those mapping to sexual chromosomes, SNPs with a genotyping rate lower than 90% or those below a minor allele frequency threshold of 0.05 were removed. To avoid departures from Hardy–Weinberg proportions due to genotyping errors, SNPs that did not passtheHardy–Weinberg test for *p* ≤ 0.001 were removed as well. A total of 46,977 SNPs located on the 26 ovine autosomes passed the quality control for the whole sample analyzed.

### 2.2. Population Structure Analyses

The program PLINK V 1.9 [23] was used to compute the between individuals distance matrix (complimentary to the between individuals identity-by-state matrix). The betweenindividuals relationships were summarized performing principal component analysis (PCA) on such a distance matrix using Proc Factor of SAS/STAT software package 9.6 (SAS Institute Inc., Cary, NC, USA, 2016). Eigenvectors computed for each individual via PCA were used to construct dispersion plots.

### 2.3. Detection of Runs of Homozygosity

ROH were identified using the approach implemented in PLINK V 1.9 [23]. Sliding windows of 50 SNPs were fitted to identify homozygous stretches. A maximum of two SNPs with a missing genotype, one heterozygote position, and a maximum gap of 1000 kb were allowed per ROH window. This approach has increased the power for detection of truly autozygous segments even in scenarios of long ROH [24]. A given SNP was considered to be part of a ROH if present in at least 5% of the 50 SNP sliding windows. PLINK offers summary statistics for the maximum (UNION) and minimum (CONSENSUS; i.e., those ROH segments shared by all individuals on which a given ROH was identified) length of a given ROH over a population. Our focus was on shorter ROH. To minimize the effect of recent demographic events giving long ROH segments and the lower ability of PLINK to identify short ROH when medium-density chips are analyzed [8], the consensus (shorter) ROH segments identified were used as a reference for subsequent analyses. In other words, we selected the shorter ROH segments in two steps: first, we identitied ROH across individuals; and later we defined the shorter segments shared by all individuals in which a given ROH was identified.

### 2.4. Homozygosity-by-Descent Analyses

Inbreeding was computed using the program ZooRoH [16]. ZooRoH implements a hidden Markov model (HMM) accounting for allele frequencies, genetic distances, genotyping error rates, and sequences read counts to determine the probability of each locus being HBD. The software needs to split the genomes analyzed into an arbitrary number (K) of HBD classes and one non-HBD class. A total of 11 different HBD classes were fitted according to the number of generations (G) separating the HBD segment from the most recent common ancestor [25]. The HBD classes fitted were G2, G4, G8, G32, …, and G2048. The non-HBD class was fit to G2048 as well [25]. In the HMM framework, ZooROH uses a forward-backward algorithm to compute the probability at each marker position to belong to each of the different K classes by integrating over all possible sequences of segments. This implied the use of a square identity matrix of order K as a transition matrix defining the allowable HBD state changes. We used default values for any other parameters needed for the ZooROH analyses. The model implemented in ZooROH assumes random mating and deviations from panmixia may bias results. However, the Djallonké sheep breeding scenario fits well to this assumption: unsupervised matings are the rule, no selection programs exist, and a strong gene flow exists between neighboring populations [4,5,7,18,19,20].

ZooROH uses estimates of recombination on both paths (paternal and maternal) to a common ancestor to define the distribution of HBD segments in each of the predefined K ancestry classes. If no recombination occurs, the rate of ancestry change (R), which defines the distribution of segments in each of the K classes, is R = 2G: the smaller the value of R, the more recent the segments. Segments classified up to generation G64 were considered attributable to recent inbreeding [25]. Segments classified into G1024 and G2048 were considered ancient inbreeding. Accepting a generation interval of three years in sheep [26], 1024 generations roughly coincide with the time in which sheep reached West Africa (3700 yBP) [2].

HBD levels of a given class K and their addition over the 11 HBD classes fitted informed on the individual inbreeding (recent, intermediate, ancient or total). Note that these values can be interpreted as inbreeding coefficients estimated with respect to different base populations separated from the individual generations ago [27]. This can be used to assess individual variation in generations contributing to autozygosity.

### 2.5. Candidate Homozygous Segments and Enrichment and Functional Annotation Analyses

Candidate genomic regions were defined starting from HBD segments of individuals having segments assigned to G1024 or G2048 classes. The overlap of these HBD segments with the consensus ROH identified using PLINK V 1.9 [23] was assessed using the intersect Bed function of the BedTools software [28]. The upper and lower bounds of these overlaps were considered candidate regions for subsequent analyses.

Protein-coding genes found within the candidate regions were retrieved from the Ensembl Genes 91 database, based on the Oar v3.1 ovine reference genome (http://www.livestockgenomics.csiro.au/sheep/oar3.1.php) using the BioMart tool [29]. All the identified genes were processed using the functional annotation tool implemented in DAVID Bioinformatics resources 6.8 [30] to determine enriched functional terms. An enrichment score of 1.3, which is equivalent to the Fisher exact test *p*-value of 0.05, was used as a threshold to define the significantly enriched functional terms in comparison to the whole bovine reference genome background. Relationships among genomic features in different chromosome positions were represented using the software package shinyCircos [31].

## 3. Results

PCA identified three different factors with eigenvalue> 1 explaining a total of 22.5% of the genetic variance. Factor 1 (eigenvalue = 26.7) explained 14.5% of the genetic variance, factor 2 (eigenvalue = 8.7) explained 4.7%, and factor 3 (eigenvalue = 6.1) explained 3.3% of the variance. Figure 1 summarizes the between individuals genetic relationships and illustrates a weak genetic structuring in data. Note that no separation between individuals was found according to sampling area (Mangodara or Dédougou).

### 3.1. Genomic Homozygosity Distribution

Using PLINK, a total of 510 ROH segments was identified on the genome of 127 individuals (69% of the samples), seven of them sampled in Dedougou. ROH segments could be summarized into 124 different ROH. Details on these 124 ROH are given in Appendix A. Size of consensus ROH varied from 1 SNP (six different ROH) to 10.1 Mb (including a maximum of 207 SNPs). Figure 2 illustrates the ROH distribution pattern. In Figure 2A, most individuals (87) clustered close to the origin of coordinates because they harbored less than five ROH. In these 87 individuals, ROH covered, on average, a total length of 10.37 Mb. Figure 2B informs on the distribution of ROH segments according to their size. Most ROH identified (239; 37.3% of the total) had a size lower than 5 Mb and only 12 of them were lower than 2 Mb (not shown). The 15–20 Mb and the >20 Mb ROH classes only gathered 4% and 5% of the ROH segments, respectively. Chromosomes 3, 2, 23, 9, 5, and 10 harbored more than 50 ROH segments each, with chromosomes 2, 5, and 23 harboring the longer ROH segments (Figure 3). Analyses could not identify ROH on eight ovine chromosomes (11, 13, 14, 16, 17, 19, 24, and 26). Four chromosomes (23, 5, 3, and 2) gathered 52% of different ROH listed in Appendix A.

### 3.2. Homozygosity-by-Descent Analyses

ZooRoH analyses informed that most of the genome segments identified (88.7%) were non-HBD. Most HBD segments identified across individuals were assigned to the intermediate inbreeding classes (G128, G256, and G512) accounting for 64.9% of the total. HBD classes including recent inbreeding (G64 and below) accounted for very few segments (0.3% of the total). G1024 and G2048 HBD classes accounted for 24.9% and 9.9% of the total, respectively. HBD fragments were identified on 119 individuals only. Fifty-eight of them did not harbor HBD segments assigned to either G1024 or G2018 classes. Average inbreeding of these 58individuals was 0.2% while that of the remaining 61 was 25.3%. Most of the inbreeding of the latter was of intermediate origin (16.6%) while that of ancient origin (G1024 and G2048) was 8.63% (Figure 4).

Details of the 32,968 HBD segments identified across individuals and chromosomes are listed in Appendix A. Only 2801 HBD fragments (8.5% of the total) were longer than 5 Mb, 21,384 (65%) of them shorter than 2 Mb, and 7591 (23%) shorter than 1 Mb. In any case, the HBD fragments identified by ZooROH covered about 698 Mb of the genome of the individuals typed on average (over 26% of the sheep genome).

The time (in generations) in which HBD fragments originated was assessed considering the mean value of the 11 possible G classes to which it could be assigned and varied from 4 to 9. Most fragments were, on average, assigned to the 8th (16,595) and 9th (10,191) classes (out of 11). These 26,786 HBD segments were kept for further analyses to ensure that these genomic areas corresponded to ancient evolutionary processes.

### 3.3. ROH-HBD Intersections and Identification of Functional Candidate Genes

The overlap between genomic ROH identified using PLINK and those HBD fragments with the higher generation class assignment gave 108 candidate regions on 17 different ovine chromosomes (Appendix A). Ovine chromosomes (OARs)2, 3, 5, and 23 gathered 12 or more candidate regions each (53 in total). The other chromosomes gathered from 1 (OAR22) to 8 (OAR10) candidate regions. Candidate regions involved a total of 351.5 Mb and 926 different HBD segments.

Gene-annotation enrichment analysis allowed to identify a total of 1688 potential candidate ovine genes (1974 different transcripts) on these 108 candidate regions. The full list of these candidate ovine genes, including their identification, description, and location, is given in Appendix A. Functional annotation conducted on these genes allowed to identify 61 different functional term clusters (Appendix A). However, only six of them were statistically significantly enriched (enrichment score higher than 1.3; Table 1 and Table 2). A description of the genes involved in the definition of these six functional clusters is given in Table 2 and the relationships between genes within the functional cluster are illustrated in Figure 5.

Functional cluster 1 (enrichment score = 3.084) is formed by nine genes located in close vicinity on OAR23 belonging to serpin (serine protease inhibitors) family. 

Functional cluster 2 (enrichment score = 2.024) included five genes encoding lysozymes located on the same locus on OAR3. 

Functional cluster 3 (enrichment score = 2.011) included eight genes belonging to the scavenger receptor genes family. 

Functional cluster 4 (enrichment score = 2.008) included five genes, mainly located on OAR3, coding proteins of the cadherin binding domain.

Functional cluster 5 (enrichment score = 1.563) included 20 genes located on 11 chromosomes linked to glycine N-acyltransferase activity. 

Finally, Functional cluster6 (enrichment score = 1.419) included three genes located in close vicinity on OAR3 encoding guanine nucleotide exchange factors (GEFs).

## 4. Discussion

The same Djallonké population analyzed here was assessed before for population structure using different markers and statistical methods [7,18,19]. In all cases, the conclusion was that the population was poorly structured. Only a few individuals appear separated on the Y-axis of Figure 1. The weak structure identified on Figure 1 can be due to unexpected matings between close relatives [18] as unsupervised matings are the rule in the traditional sheep management system of Burkina Faso4. As an additional verification, the distribution of ROH between the main Djallonké group and that of the individuals separated on the Y-Axis was analyzed using Mantel–Haenszel Chi-squared test (Proc Freq of SAS/STAT software package 9.6). Analysis informed that the distribution of ROH between two groups did not depart from random expectation (*p* = 0.414).

### 4.1. Homozygosity and Autozygosity in Djallonké Sheep

There were substantial differences in the number and size of homozygous segments detected by PLINK and ZooROH. This is likely due to the performance differences of the algorithms implemented in these software. The sliding window method implemented in PLINK has been shown to give biased results overestimating the numbers of short ROH segments (between 1 and 4 Mb) when analyzing medium-density SNP chips [13,32]. Therefore, some of the segments identified by PLINK may be actually longer and their informative ability to characterize ancient genomic evolutionary processes questioned. However, the Viterbi algorithm implemented in ZooROH16 can reliably detect homozygous segments shorter than 1 Mb using that kind of chip [25]. In our research, ZooROH identified a high number of short fragments covering, as a whole, a substantial part of the sheep genome (Appendix A; about 698 Mb per individual, on average; over 26% of the genome). This makes the use of ZooROH results unsuitable only to ascertain the impact of adaptation to the West African environment on the sheep genome. The contrast between results obtained using PLINK and ZooROH allowed to narrow down the search genomic areas.

In any case, the combined use of the information provided by observational methods and that of ZooROH has a clear justification. HBD lengths are inversely related to the time of origin [16,17,27]. However, very small HBD segments with strong directional selection pressure originated deep in the past can be added to a larger, more recent, ROH due to founder events or bottlenecks therefore changing the distribution of ancient HBD segments [33,34,35,36]. Therefore, it may not be advised using as markers for selection events short HBD segments only. Although ancient HBD segments may be found in different populations in ROH of different sizes depending on the specific demographic history of the considered population [33,34], we here analyze a single population and our results are not likely to be biased by different demographic events.

### 4.2. Biological Importance of the Functional Clusters Identified

The six functional term clusters identified via enrichment are involved in signaling pathways putatively associated with adaptation to challenging environments: functional cluster 1 is involved in homeostasis and coagulation; functional clusters 2, 3, and 6 are associated to innate immunity, defense against infections, and cell white blood cells proliferation and migration, respectively; functional cluster 4 is involved in parasite resistance; functional cluster 5 is involved in response to stress.

Functional cluster 1 includes genes coding serpins which are a broadly distributed family of protease inhibitors that use a conformational change to inhibit target enzymes. They are central in controlling many important proteolytic cascades, including the mammalian coagulation pathways [37].

Functional cluster 2 is formed by genes encoding lysozymes. Lysozymes participate in host defense against bacterial infection with a primarily bacteriolytic function in association with the monocyte-macrophage system and the enhancement of the activity of immunoagents. They also work in the stomach of ruminants to digest bacterial cell walls and are presumed to account for much of the functional adaptation [38].

The genes included in functional cluster 3 code scavenger receptors, a ‘superfamily’ of membrane-bound receptors known to bind to a variety of ligands including endogenous proteins and pathogens with biological functions such as clearance of modified lipoproteins and pathogens. Scavenger receptors regulate patho-physiological states including pathogen infections and immune surveillance [39].

Functional cluster 4 is the “Cadherin Cluster”. Cadherins comprise a large family of Ca^2+^-dependent cell–cell adhesion molecules. E-Cadherin, a prototypical member of this family, is a transmembrane protein that forms the adherens junction between epithelial cells. This implies a function on resistance to parasites including *Haemonchus contortus* [40].

Genes forming functional cluster 5 are involved in the activity of the glycine N-acyltransferase promoting lysine acetylation. This is a major post-translational modification of proteins which regulates many physiological processes such as metabolism, cell migration, aging, and inflammation. They have been identified as signaling molecules that regulate functions like the perception of pain and body temperature and also have anti-inflammatory properties [41].

The genes included in functional cluster 6 are guanine exchanging factors, proteins stimulating the exchange of guanosine 5’-diphosphate for guanosine-5’-triphosphate to activate many downstream targets, or effector proteins. They are involved in lymphocyte development and signaling and T-cell development and signaling downstream of T-cell receptor complexes [42].

### 4.3. Consistency with Previous Analyses

A genomic scan of the current Djallonké sheep sample was recently carried out [7] for the identification of selection sweeps to obtain new insights on sheep adaptation to harsh environments. Since that and the current studies cannot be considered independent, a joint discussion of the current findings and those previously reported is advisable. In our previous report [7], three complementary extended haplotype homozygosity-based statistics (iHS, XP-EHH, and nSL) allowed to identify alleles under selection with complete or incomplete fixation were used to define candidate selection sweep regions. The comparison of the information provided by these tests allowed to identify 207 candidate selection sweep regions comprising about 35 Mb. Fifty-four of these selective sweeps overlapped with a total of 44 candidate regions defined here (Appendix A). Overlap occurred on 14 different ovine chromosomes, with OAR23 having overlapped with 10 different selection sweeps.

More importantly, up to 20 genes included in four out of six functional clusters identified in the current research (Table 2; Appendix A) were located in the genomic regions in which the current candidate regions and selection sweeps previously reported overlap (genes *LYZ1*, ENSOARG00000020417, *LYSB*, *LYZ2*, and *LYZ* of functional cluster 2; genes *PTPRR*, *PTPRB*, *PTPN2*, and ENSOARG00000009449 of functional cluster 3; gene *TMPRSS15* of functional cluster 4; genes *FBXW2*, *NOL10*, *VPS41*, *HERC1*, *TBC1D31*, *WDFY2*, and *NBEA* functional cluster 5; and genes *ERAP1*, *ERAP2* and *LNPEP* of functional cluster6).

According to the literature, overlap between selection signatures and ROH segments is usually poor [9,10]. Both in sheep and goat, it has been reported that ROH tend to map to selective sweeps in highly selected breeds [9,12]. Djallonké sheep analyzed here are not subject to selection programs for production traits. Therefore, the moderate overlap identified can be considered a confirmation of the existence of processes of natural selection for adaptation in our population (Appendix A). The previous work identified three different functional clusters including genes coding proteins mainly involved in pathways related to metabolic response to stress and, to a lesser extent, in immune response [7]. The genes located in the overlapping genomic areas belong to functional clusters associated to immunity, infections and parasite resistance, and response to stress. Such functions fit well with the hypothesis suggesting that domestic small ruminants had to adapt to a challenging environment before entering into humid Sub-Saharan West Africa [3].

## 5. Conclusions

This research proposes a new approach to use ROH segments as markers for the identification of genomic areas potentially subjected to non-recent evolutionary selective events. Their combined use with very small HBD segments with strong directional selection pressure originated deep in the past has been applied to obtain information on the effect on the sheep genome of human-mediated selection for adaptation to a harsh environment. The analysis of the Djallonké sheep genome has allowed the identification of genomic signatures putatively linked to the process of sheep adaptation to the harsh environment of the hot-humid, trypanosome challenged, West Africa. The current analysis confirms the importance of genomic areas associated to immunity, disease resistance, and response to stress for adaptation to harsh environments.

## Figures and Tables

**Figure 1 animals-10-01178-f001:**
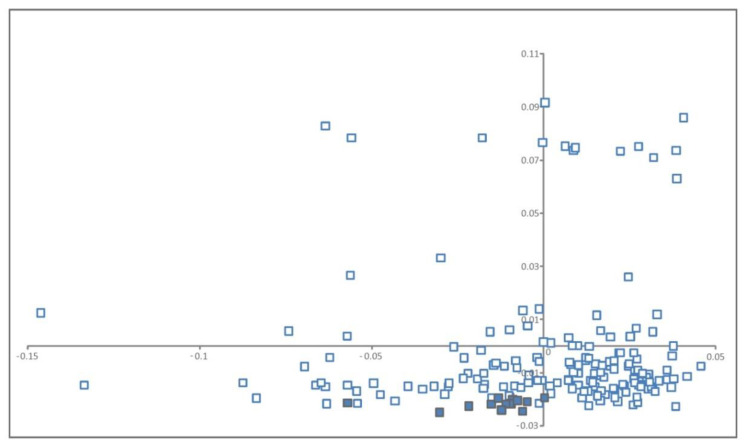
Dispersion of the individual Djallonké sheep genotypes on the bidimensional space formed by the two first factors retained after computation of principal component analysis (PCA) on the between individuals distance matrix. Factor 1, on the X-axis explained 14.5% of the variance. Factor 2, on the Y-axis, explained 4.7% of the variance. Individuals sampled in the surroundings of Mangodara are in open squares while those sampled in Dédougou are in solid squares.

**Figure 2 animals-10-01178-f002:**
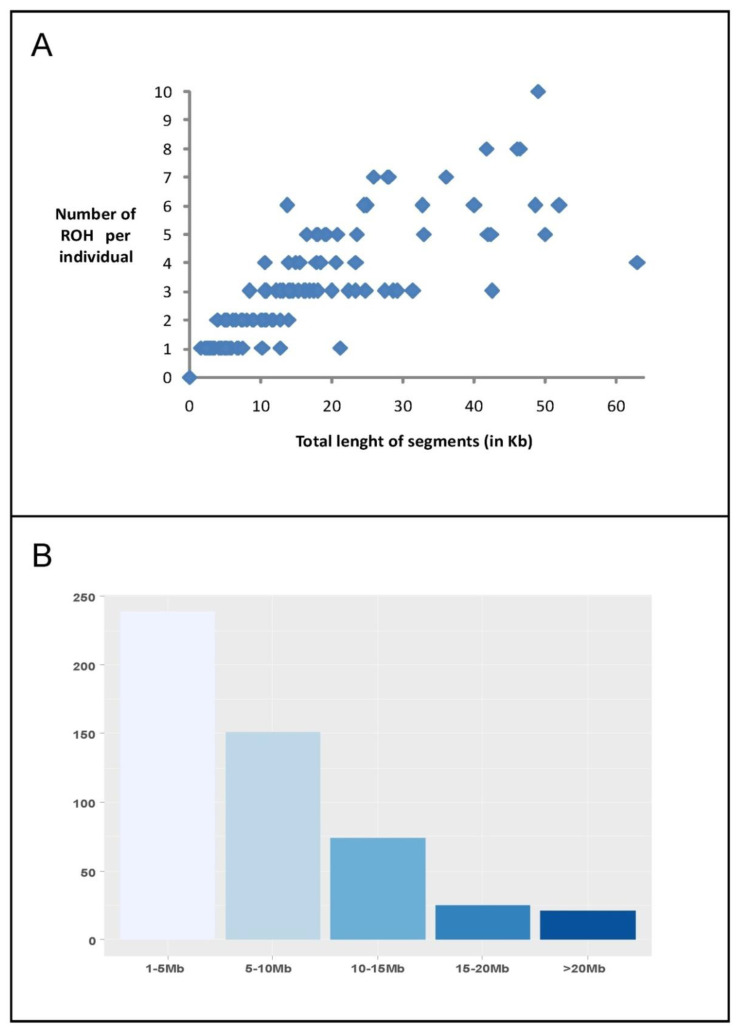
Summary of the identification runs of homozygosity (ROH) carried out using the program PLINK. (**A**) The ROH complement (total length and number) per individual in Djallonké sheep. (**B**) The frequency of ROH according to their length.

**Figure 3 animals-10-01178-f003:**
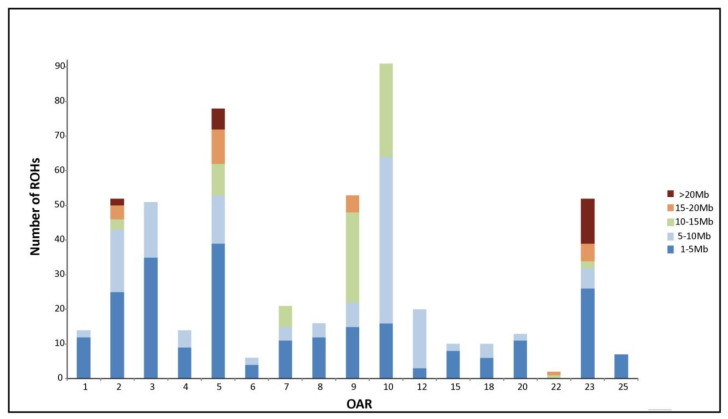
Runs of homozygosity distribution and coverage for each autosome in Djallonké sheep. The barplot shows the number of runs of homozygosity in different length classes: 1–5 Mb, 5–10 Mb, 10–15 Mb, 15–20 Mb, and >20 Mb.

**Figure 4 animals-10-01178-f004:**
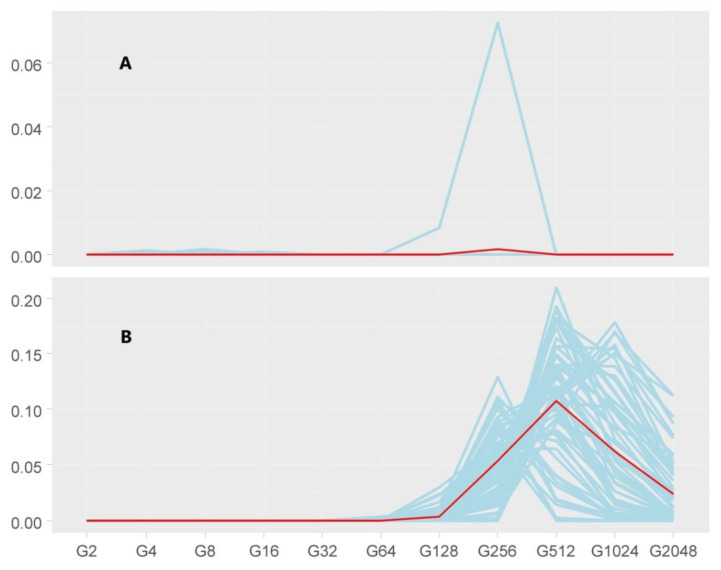
Genomic inbreeding coefficients estimated with respect to different base populations (in 11 generation classes from G2 to G2048) estimated as the probability of a given genomic segment belonging to any of the 11 homozygosity-by-descent (HBD) classes fitted. (**A**) The variation in genomic inbreeding of individuals with no HBD segments assigned to G1024 or G2048. (**B**) The same variation for the individuals harboring HBD segments kept for enrichment analyses. Dark lines are the average values for each subpopulation. Note that (**A**) and (**B**) have different ranges on the Y-axis.

**Figure 5 animals-10-01178-f005:**
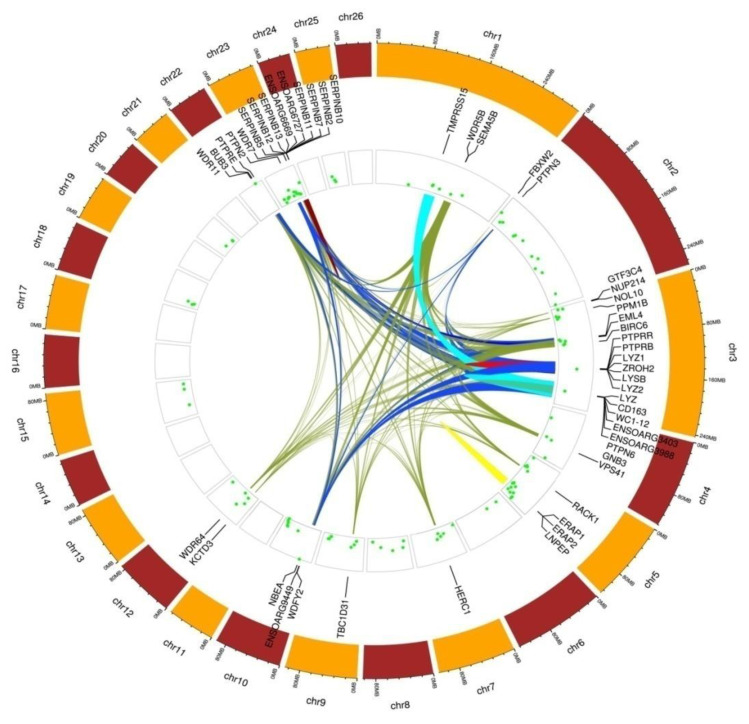
Circular map summarizing information on functional clusters identified in Djallonké sheep. Chromosomes are shown in the outermost circle. The innermost circles show the distribution of the candidate genomic areas identified within a chromosome. Differences in height among candidate areas inform on the number of SNPs defining them. Candidate genes forming the significantly enriched functional term clusters are indicated beside their genomic localization. At the center of the map, links among candidate genes belonging to the same functional cluster are illustrated using the same color (cluster 1 in brown on OAR23; cluster 2 in red on OAR3; cluster 3 in dark blue; cluster 4 in light blue; cluster 5 in green; cluster 6 in yellow on OAR5).

**Table 1 animals-10-01178-t001:** Significantly enriched functional term clusters (enrichment score higher than 1.3) for genes identified within candidate genomic regions following DAVID analysis.

Functional Cluster (Enrichment Score)	Term Description		Count	*p*-Value
**Cluster1 (3.084)**	INTERPRO	IPR000215:Serpin family	11	7.49 × 10^−4^
	INTERPRO	IPR023796:Serpin domain	11	7.49 × 10^−4^
	SMART	SM00093:SERPIN	11	8.29 × 10^−4^
	INTERPRO	IPR023795:Protease inhibitor I4, serpin, conserved site	10	9.96 × 10^−4^
**Cluster2 (2.024)**	INTERPRO	IPR019799:Glycoside hydrolase, family 22, conserved site	5	0.005281
	INTERPRO	IPR000974:Glycoside hydrolase, family 22, lysozyme	5	0.007487
	INTERPRO	IPR001916:Glycoside hydrolase, family 22	5	0.010224
	SMART	SM00263:LYZ1	5	0.010761
	GOTERM_MF_DIRECT	GO:0003796~lysozyme activity	5	0.012172
	INTERPRO	IPR023346:Lysozyme-like domain	5	0.013533
**Cluster3 (2.011)**	INTERPRO	IPR001190:Speract/scavenger receptor	8	0.004788
	SMART	SM00202:SR	8	0.005171
	INTERPRO	IPR017448:Speract/scavenger receptor-related	8	0.008159
	GOTERM_MF_DIRECT	GO:0005044~scavenger receptor activity	8	0.044628
**Cluster4 (2.008)**	INTERPRO	IPR009122:Desmosomal cadherin	7	7.62 × 10^−7^
	INTERPRO	IPR027397:Catenin binding domain	7	0.005220
	INTERPRO	IPR000233:Cadherin, cytoplasmic domain	6	0.019394
	INTERPRO	IPR002126:Cadherin	10	0.036299
	INTERPRO	IPR015919:Cadherin-like	10	0.039529
	SMART	SM00112:CA	9	0.070844
	GOTERM_BP_DIRECT	GO:0007156~homophilic cell adhesion via plasma membrane adhesion molecules	10	0.094881
	INTERPRO	IPR020894:Cadherin conserved site	8	0.116413
**Cluster5 (1.563)**	INTERPRO	IPR010313:Glycine N-acyltransferase	3	0.026686
	INTERPRO	IPR015938:Glycine N-acyltransferase, N-terminal	3	0.02668622
	INTERPRO	IPR013652:Glycine N-acyltransferase, C-terminal	3	0.026686
	GOTERM_MF_DIRECT	GO:0047961~glycine N-acyltransferase activity	3	0.029378
**Cluster6 (1.419)**	INTERPRO	IPR000651:Ras-like guanine nucleotide exchange factor, N-terminal	6	0.016082
	SMART	SM00229:RasGEFN	5	0.028570
	SMART	SM00147:RasGEF	6	0.051153
	INTERPRO	IPR001895:Guanine-nucleotide dissociation stimulator CDC25	6	0.054995
	INTERPRO	IPR023578:Ras guanine nucleotide exchange factor, domain	6	0.061954

**Table 2 animals-10-01178-t002:** Description of the sheep genes included in the six statistically significant functional clusters displaying enrichment score higher than 1.3.In addition to the gene name and description, the following information is provided: the identification of the gene retrieved from the Ensembl Genes 91 database, the ovine chromosome (OAR) on which the gene is located, and the positions (in bp) of start and end of the gene within the chromosome.

Functional Cluster (Enrichment Score)	Gene Name	Description	EnsemblID	OAR	Gene Start	Gene End
**Cluster1 (3.084)**	*SERPINB5*	serpin family B member 5 [Source:HGNC Symbol;Acc:HGNC:8949]	ENSOARG00000006391	23	61974470	61989454
	*SERPINB12*	serpin family B member 12 [Source:HGNC Symbol;Acc:HGNC:14220]	ENSOARG00000006441	23	62028783	62041541
	*SERPINB13*	serpin family B member 13 [Source:HGNC Symbol;Acc:HGNC:8944]	ENSOARG00000006513	23	62045722	62057218
	ENSOARG00000006669		ENSOARG00000006669	23	62100351	62105829
	ENSOARG00000006727	serpin B4-like [Source:NCBI gene;Acc:101104114]	ENSOARG00000006727	23	62117011	62125799
	*SERPINB11*	serpin family B member 11 (gene/pseudogene) [Source:HGNC Symbol;Acc:HGNC:14221]	ENSOARG00000006806	23	62162337	62177361
	*SERPINB7*	serpin family B member 7 [Source:HGNC Symbol;Acc:HGNC:13902]	ENSOARG00000006880	23	62189501	62220423
	*SERPINB2*	plasminogen activator inhibitor 2 [Source:NCBI gene;Acc:101105044]	ENSOARG00000006889	23	62287023	62298724
	*SERPINB10*	serpin family B member 10 [Source:HGNC Symbol;Acc:HGNC:8942]	ENSOARG00000006971	23	62308486	62330628
**Cluster2 (2.024)**	*LYZ1*	Ovisarieslysozyme C-1-like (LOC443320), mRNA. [Source:RefSeq mRNA;Acc:NM_001308588]	ENSOARG00000020393	3	150160411	150294070
	ENSOARG00000020417	lysozyme C, tracheal isozyme [Source:NCBI gene;Acc:101102969]	ENSOARG00000020417	3	150225121	150229639
	*LYSB*	lysozyme C, intestinal isozyme [Source:NCBI gene;Acc:101103222]	ENSOARG00000020429	3	150266228	150270946
	*LYZ2*		ENSOARG00000020439	3	150313875	150318870
	*LYZ*	lysozyme [Source:NCBI gene;Acc:100049062]	ENSOARG00000020515	3	150434369	150439510
**Cluster3 (2.011)**	ENSOARG00000009449		ENSOARG00000009449	10	21972591	22033480
	*PTPN3*	protein tyrosine phosphatase non-receptor type 3 [Source:HGNC Symbol;Acc:HGNC:9655]	ENSOARG00000006968	2	13790622	13898273
	*PTPRE*	protein tyrosine phosphatase receptor type E [Source:HGNC Symbol;Acc:HGNC:9669]	ENSOARG00000014124	22	46388779	46432003
	*PTPN2*	protein tyrosine phosphatase non-receptor type 2 [Source:HGNC Symbol;Acc:HGNC:9650]	ENSOARG00000001930	23	43434719	43479972
	*PTPN6*	protein tyrosine phosphatase non-receptor type 6 [Source:HGNC Symbol;Acc:HGNC:9658]	ENSOARG00000005032	3	207454996	207463860
	*PPM1B*	protein phosphatase, Mg2+/Mn2+ dependent 1B [Source:NCBI gene;Acc:101112467]	ENSOARG00000006389	3	79989814	80014651
	*PTPRR*	protein tyrosine phosphatase receptor type R [Source:HGNC Symbol;Acc:HGNC:9680]	ENSOARG00000020111	3	148667786	148945673
	*PTPRB*	protein tyrosine phosphatase receptor type B [Source:HGNC Symbol;Acc:HGNC:9665]	ENSOARG00000020158	3	148947442	149071453
**Cluster4 (2.008)**	*TMPRSS15*	transmembrane serine protease 15 [Source:HGNC Symbol;Acc:HGNC:9490]	ENSOARG00000015635	1	136961183	137107011
	*CD163*	CD163 molecule [Source:HGNC Symbol;Acc:HGNC:1631]	ENSOARG00000002862	3	206525820	206562443
	*WC1-12*		ENSOARG00000003251	3	206712983	206746704
	ENSOARG00000003403		ENSOARG00000003403	3	206786488	206816410
	ENSOARG00000003988		ENSOARG00000003988	3	207068439	207075844
**Cluster5 (1.563)**	*WDR5B*	WD repeat domain 5B [Source:HGNC Symbol;Acc:HGNC:17826]	ENSOARG00000001983	1	185127233	185128225
	*SEMA5B*	semaphorin 5B [Source:HGNC Symbol;Acc:HGNC:10737]	ENSOARG00000020173	1	185521338	185559506
	*WDFY2*	WD repeat and FYVE domain containing 2 [Source:HGNC Symbol;Acc:HGNC:20482]	ENSOARG00000008939	10	21328931	21506011
	*NBEA*	neurobeachin [Source:HGNC Symbol;Acc:HGNC:7648]	ENSOARG00000010627	10	26007917	26592574
	*WDR64*	WD repeat domain 64 [Source:HGNC Symbol;Acc:HGNC:26570]	ENSOARG00000007794	12	33167064	33301577
	*KCTD3*		ENSOARG00000010181	12	16935183	16992237
	*FBXW2*	F-box and WD repeat domain containing 2 [Source:HGNC Symbol;Acc:HGNC:13608]	ENSOARG00000005690	2	2693449	2722938
	*WDR11*	WD repeat domain 11 [Source:HGNC Symbol;Acc:HGNC:13831]	ENSOARG00000004701	22	39787927	39845136
	*BUB3*		ENSOARG00000009467	22	41838447	41848859
	*WDR7*	WD repeat domain 7 [Source:HGNCSymbol;Acc:HGNC:13490]	ENSOARG00000005109	23	56218845	56550504
	*GNB3*	G protein subunit beta 3 [Source:HGNC Symbol;Acc:HGNC:4400]	ENSOARG00000005728	3	207555357	207560893
	*GTF3C4*	general transcription factor IIIC subunit 4 [Source:HGNC Symbol;Acc:HGNC:4667]	ENSOARG00000005793	3	4112816	4127921
	*NUP214*	nucleoporin 214 [Source:HGNC Symbol;Acc:HGNC:8064]	ENSOARG00000007056	3	5440880	5534974
	*EML4*	EMAP like 4 [Source:HGNC Symbol;Acc:HGNC:1316]	ENSOARG00000007545	3	81676763	81769277
	*BIRC6*	baculoviral IAP repeat containing 6 [Source:HGNC Symbol;Acc:HGNC:13516]	ENSOARG00000010515	3	91252791	91461884
	*NOL10*	nucleolar protein 10 [Source:HGNC Symbol;Acc:HGNC:25862]	ENSOARG00000015507	3	19613346	19705109
	*VPS41*	VPS41 subunit of HOPS complex [Source:HGNC Symbol;Acc:HGNC:12713]	ENSOARG00000017742	4	82007947	82218383
	*RACK1*	receptor for activated C kinase 1 [Source:NCBI gene;Acc:100137070]	ENSOARG00000007288	5	37884620	37892060
	*HERC1*	HECT and RLD domain containing E3 ubiquitin protein ligase family member 1 [Source:HGNC Symbol;Acc:HGNC:4867]	ENSOARG00000020775	7	43237435	43432873
	*TBC1D31*	TBC1 domain family member 31 [Source:HGNC Symbol;Acc:HGNC:30888]	ENSOARG00000010832	9	29487020	29549424
**Cluster6 (1.419)**	*ERAP1*	endoplasmic reticulum aminopeptidase 1 [Source:HGNC Symbol;Acc:HGNC:18173]	ENSOARG00000017807	5	93487749	93518183
	*ERAP2*	endoplasmic reticulum aminopeptidase 2 [Source:HGNC Symbol;Acc:HGNC:29499]	ENSOARG00000017926	5	93629581	93674652
	*LNPEP*	leucyl and cystinylaminopeptidase [Source:HGNC Symbol;Acc:HGNC:6656]	ENSOARG00000017994	5	93687872	93788740

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
