# Peer review of "Ancient Homozygosity Segments in West African Djallonké Sheep Inform on the Genomic Impact of Livestock Adaptation to the Environment"

_animals, 2020, doi:10.3390/ani10071178_

Round 1
Reviewer 1 Report
This manuscript describes an analyses using runs of homozygosity (ROH) and homozygosity-by-descent (HBD) to examine genomic impact of livestock adaptation to the environment in Djallonke sheep. It is important to understand the origins of important traits in livestock that allow animals to resist disease and stress. The paper is well-written with only minor comments included below for revision.
- Figure 3 font size should be increased.
- Citation numbering appears to be off by 1: L315 (should it include citation 33?), L326 (should be 34?), L331 (should be 35?), L336 (should be 36?), L340 (should be 37?), L345 (should be 38?), L349 (should be 39?).
- L362: “…four out of six…”
- L364: fix “reported7”
- Fix formatting: L451, L482 (capitals needed).
Author Response
This manuscript describes an analyses using runs of homozygosity (ROH) and homozygosity-by-descent(HBD) to examine genomic impact of livestock adaptation to the environment in Djallonke sheep. It is important to understand the origins of important traits in livestock that allow animals to resist disease and stress. The paper is well-written with only minor comments included below for revision.
ANSWER: Thank you for your words
Point1: Figure 3 font size should be increased.
ANSWER: done as requested.
Point2: Citation numbering appears to be off by 1: L315 (should it include citation 33?), L326 (should be 34?), L331 (should be 35?), L336 (should be 36?), L340 (should be 37?), L345 (should be 38?), L349 (should be 39?).
ANSWER: The cites form 31 to 34 are included before in L312 “….HBD segments [31-34].” Citations [31-32] were cited again in L315 due to discussion needs. From there onwards citations are correct. Please note that due to the inclusion of two new citations, as requested by Reviewer #2, previous citations [31-34] and [31-32] are now [33-36] and [33-34], respectively (L318-321).
Point3: L362: “…four out of six…”
ANSWER: corrected as requested. Thank you.
Point4: L364: fix “reported7”
ANSWER: corrected as requested. Thank you.
Point5: Fix formatting: L451, L482 (capitals needed).
ANSWER: corrected as requested. Thank you.
Reviewer 2 Report
This manuscript proposes a combined approach to use ROH and HBD segments to obtain evolutionary selective events on the Djallonké sheep genome for adaptation to a harsh environment in Sub-Saharan West Africa. The research work presented in a manner of logical, innovative and integral form. However, some questions need to be interpreted or replenished.
- In the ‘Data available’ section, the protocol for DNA extraction from blood and hair roots should be written clearly although the samples were obtained from previous projects.
- It is noted in the discussion part, which described the relationship between the harsh environment and the genomic areas with a large chunk of text on the functional genomic areas, but provided weak explanations of the significance of the environment. andI suggest authors try to dig deeper into the role of challenging environment of humid Sub-Saharan West Africa, and discuss intensively the relationships about the genomic areas uncovered in this study.
- This manuscript took ROH as one of the study objects, and analyzed non-recent evolutionary selective events under the ROH and population structure, selection strength, mating system and other factors related to ROH, but the effect of population size on ROH has not been evaluated in data processing since population size is one of the factors which impact ROH on the genome.
- It is strongly suggested to carefully proofreading, grammars, spellings, sentence usages and punctuation mistakes. For example, the spelling error in the title of Figure 2-Plot A: ‘lenght’.
Author Response
Point1:This manuscript proposes a combined approach to use ROH and HBD segments to obtain evolutionary selective events on the Djallonké sheep genome for adaptation to a harsh environment in Sub-Saharan West Africa. The research work presented in a manner of logical, innovative and integral form. However, some questions need to be interpreted or replenished.
ANSWER: Thank you for your words. We have accepted your suggestions and modified our manuscript accordingly. We hope you can consider this revised version of our manuscript suitable for publication in Animals.
Point2:In the ‘Data available’ section, the protocol for DNA extraction from blood and hair roots should be written clearly although the samples were obtained from previous projects.
ANSWER: This information is now in L99-100. A new citation (22. Sambrook, J.; Fritsch, E.F.; Maniatis, T. Molecular Cloning, A Laboratory Manual, 2nd ed.; Cold Spring Harbor LaboratoryPress: Cold Spring Harbor, USA, 1989; pp. 1546.). Subsequent references renumbered accordingly.
Point3: It is noted in the discussion part, which described the relationship between the harsh environment and the genomic areas with a large chunk of text on the functional genomic areas, but provided weak explanations of the significance of the environment. And I suggest authors try to dig deeper into the role of challenging environment of humid Sub-Saharan West Africa, and discuss intensively the relationships about the genomic areas uncovered in this study.
ANSWER: New information is now given in L101-109. A new citation (23. Traoré, A.; Álvarez, I.; Fernández, I.; Pérez-Pardal, L.; Kaboré, A.; Ouédraogo-Sanou, G.M.S.; Zaré, Y.; Tamboura H.H.; Goyache, F. Ascertaining gene flow patterns in livestock populations of developing countries: a case study in Burkina Faso goat. BMC Genetics 2012, 13, 35. https://doi.org/10.1186/1471-2156-13-35) has been included in the References section to inform to the readers on agroecological constraints affecting tsetse fly distribution and trypanosome challenge. Subsequent references have been renumbered accordingly.
Point4: This manuscript took ROH as one of the study objects, and analyzed non-recent evolutionary selective events under the ROH and population structure, selection strength, mating system and other factors related to ROH, but the effect of population size on ROH has not been evaluated in data processing since population size is one of the factors which impact ROH on the genome.
ANSWER: We agree with reviewer that length and distribution of ROH in the genome may be affected by effective population size. However, as explained in L150-153, this sample has not breeding constraints affecting effective population size. Furthermore, sample is considerably larger than others available in similar studies and has been shown to have negligible genetic structuring. Therefore, we consider that it is not likely that effective population size is affecting our results.
It is strongly suggested to carefully proofreading, grammars, spellings, sentence usages and punctuation mistakes. For example, the spelling error in the title of Figure 2-Plot A: ‘lenght’.
ANSWER: English grammar and typos have been carefully revised and corrections included in the text of the manuscript, when necessary.
Reviewer 3 Report
In this article the authors analyzed and identified Burkina Faso Djallonké sheep genomic areas of importance to adaptation to the harsh environments. The authors used different models for the illustration of the current findings very nicely, however the discussion section seems like introduction, mostly focused on the models used in the manuscript instead of discussing findings regarding ROH segments as markers for the identification of genomic areas especially for adaptation to a harsh environment of the Djallonké sheep.
The authors have divided the discussion into 3 parts, lacking consistency and linkages. The author should re-write the discussion section specifically to focus on the objectives and findings of the present manuscript.
Author Response
Point1:In this article the authors analyzed and identified Burkina Faso Djallonké sheep genomic areas of importance to adaptation to the harsh environments. The authors used different models for the illustration of the current findings very nicely, however the discussion section seems like introduction, mostly focused on the models used in the manuscript instead of discussing findings regarding ROH segments as markers for the identification of genomic areas especially for adaptation to a harsh environment of the Djallonké sheep.
ANSWER: Thank you for your words. We hope you can consider this revised version of our manuscript suitable for publication in Animals.
Point2:The authors have divided the discussion into 3 parts, lacking consistency and linkages. The author should re-write the discussion section specifically to focus on the objectives and findings of the present manuscript.
ANSWER: We, respectfully, considered necessary to keep the three subsections due to the following reasons: a ZooROH analyses may not be known enough for a part of the interested readers and, therefore, deserve particular attention (subsection 4.1); b the main biological implications of the analyses carried out are the main part of the Discussion section (subsection4.2); and c coincidences and differences with previous related research carried on the same population deserved attention to avoid confusions to interested readers (subsection 4.3).
Reviewer 4 Report
In this study, authors systematically analyzed the homozygous events of ancestors of Burkina Faso Djallonké (West African Dwarf) sheep. In the end, they identified multiple pathways or genes related to the humid environment in West Africa. This research provides new insights into the adaptation of sheep and other livestock to the extremely humid environment in West Africa. The manuscript is in general properly written, well-edited, and contains novel information. Several comments and suggestion were proposed as follows.
Minor comments:
1. Line 44, “,” should be “;”.
- Please add more literature about Djallonké sheep in the introduction.
- Line 110, “P” should be italic.
- I think “structuring” should be replaced with “structure”.
- Line 139, “The non-HBD class was fitted to G2048 as well.” Why ???. Please explain.
6. Line 147, The “.” in the reference should be at the end of the sentence.
- It can be found in PCA that the genetic difference between individuals in the first quadrant and the second quadrant is large. What is the reason for this?
- Line 221“Fifty-eight”, Line222 “58”, Please be consistent.
- Fig 4B shows why the inbreeding coefficient from G2 to G64 is 0, while the inbreeding coefficient of G512 is the highest and G2048 declined???
- Try to combine Table 2 into one page.
- Please unify the format of references.
Author Response
Comments and Suggestions for Authors
In this study, authors systematically analyzed the homozygous events of ancestors of Burkina Faso Djallonké (West African Dwarf) sheep. In the end, they identified multiple pathways or genes related to the humid environment in West Africa. This research provides new insights into the adaptation of sheep
and other livestock to the extremely humid environment in West Africa. The manuscript is in general properly written, well-edited, and contains novel information. Several comments and suggestion were proposed as follows.
ANSWER: Thank you for your words. We have accepted your suggestions and modified our manuscript accordingly. We hope you can consider this revised version of our manuscript suitable for publication in Animals.
Minor comments:
Point1. Line 44, “,” should be “;”.
ANSWER: corrected as requested. Thank you.
Point2.Please add more literature about Djallonké sheep in the introduction.
ANSWER: There are three additional citations on this population in the Materials and Methods section [18-20]. We have preferred to keep them in their original locations to maintain the consistency of the manuscript.
Point3.Line 110, “P” should be italic.
ANSWER: corrected as requested. Thank you.
Point4.I think “structuring” should be replaced with “structure”.
ANSWER: corrected as requested. Thank you.